# Undetectable Steganography for Language Models

**Or Zamir**                                                                     *orzamir@tauex.tau.ac.il*
*School of Computer Science*
*Tel Aviv University*

**Reviewed on OpenReview:** *https://openreview.net/forum?id=fq6aQoMSHz*

## Abstract

We introduce a cryptographic method to hide an arbitrary secret payload in the response of a Large Language Model (LLM). A secret key is required to extract the payload from the model's response, and without the key it is provably impossible to distinguish between the responses of the original LLM and the LLM that hides a payload. In particular, the quality of generated text is not affected by the payload. Our approach extends a recent result of Christ, Gunn and Zamir (2023) who introduced an undetectable watermarking scheme for LLMs.

| **Prompt:** | Write an email asking my professor Prof. Hannity to not make the final exam in Machine Learning 101 too difficult. |
|---|---|

| **Response:** | Dear Professor Hannity, |
|---|---|

I hope this email finds you well. I am writing to respectfully request that you consider the level of difficulty for the final exam in Machine Learning 101. While I am confident in my understanding of the course materials and have put in a significant amount of effort throughout the semester, I do have concerns about the potential difficulty of the final exam.

As you may recall, several students in my previous sections of Machine Learning have found the final exam to be very challenging, leading to frustration and disappointment. While I understand that the course is intended to push students to their limits and beyond, I believe that some relaxation of the latter may be in order.

I would kindly ask that you consider reducing the difficulty of the final exam or offering some additional supports or resources to help students prepare. I believe that this could enhance the learning experience

| **Hidden Payload:** | OZ *(Encoded as* 10 *bits.)* |
|---|---|

Figure 1: We asked Llama 2 to write an email urging a professor for an easy exam, intended to be sent anonymously. Nevertheless, the part of the response shown above secretly encodes the initials of the user who used the LLM. This secret payload was encoded without modifying the response distribution *at all*.

## 1 Introduction

As the applications of machine learning models generating human-like text become widespread, concerns for misuse proportionally rise. As detection of text generated by Large Language Models (LLMs) seemingly approaches infeasibility (JAML20; KSK[+]23; SKB[+]23; CBZ[+]23), the ability to *intentionally plant* a watermark in LLM-generated text becomes the most viable approach to differentiate LLM-generated from human-generated text.

A long line of works showed that a watermark can be planted in LLMs by altering the output texts (AF21; QZL[+]23; YAJK23; MZ23; KGW[+]23a). Recently, Christ, Gunn and Zamir (CGZ23) showed that a watermark can be planted in LLM-outputs *without altering the distribution* of responses. Informally, CGZ show that any LLM can be modified such that: 1) It is computationally infeasible to distinguish between the original and

the modified LLMs unless you hold a secret key, even when you are allowed to make many adaptive queries, 2) With the secret key, outputs of the modified LLM can be detected as watermarked. The importance of this notion of undetectibility is that it formally guarantees that the quality of text does not degrade (or change at all) in the process of planting the watermark.

In this work, we extend the watermarking scheme of CGZ to plant an *arbitrary payload* into the LLM response, while still maintaining the same property of undetectability. This implies, for example, that an LLM may secretly hide session-metadata (such as the user's name, the time, or the prompt used) in the response it gives. In particular, it can extend the ability of detecting text generated by the LLM to the ability of also knowing *who* used the LLM or *when* they did so. It also means that even an LLM running off-line may use the responses it gives to covertly leak internal information to those who will later be exposed to the seemingly-clean LLM responses. The main technical tool we use to transform the CGZ scheme from watermarking to embedding a message, is incorporating a dynamic variant of Error Correcting Codes with feedback.

The process of encoding a hidden message into a given channel (e.g., a picture or natural language) is called *Steganography* and was extensively studied in many domains (HvAL08; DIRR09; KJGR21; dWSK⁺22). The unique property of our setting, though, is that in the case of LLM responses the distribution of the channel is unknown. The hidden message should be recovered from the response of the LLM, without knowing what the prompt used to generate the response was. In particular, the process of decoding the message must be agnostic to the distribution of responses. Recent manuscripts (FCT⁺23; WYC⁺23; YAK23) embed messages in LLM responses, but do so by altering the distribution of responses, similar to the non-undetectable watermarking schemes. In particular, those methods affect the response distribution and may degrade the quality of generated text.

Under realistic assumptions which we verify empirically, the amount of hidden bits our scheme can encode in an LLM output is linear in the length of the response, which is asymptotically optimal. Our scheme only negligibly affects the complexity of the generation process and is thus easy and efficient to deploy. While this paper is theoretical in nature and the properties of the suggested scheme are proven rigorously, we also implemented the scheme and provide empirical examples. An open problem not addressed in this paper is that our encoding scheme is not very robust to edits of the generated text, we discuss this in length in later sections.

## 1.1 Organization of the paper

In Section 2 we give the formal definitions of the problem's setting, and state the main theorems of this paper rigorously. We also give the necessary preliminaries. In Section 3 we give a quick overview of the CGZ watermark. In Section 4 we give an high-level overview of our scheme. Then, in Section 5 we introduce a simple dynamic error correcting code with feedback, which we later use as a building block. Finally, in Section 6 we give and analyze the full scheme. In Section 7 we discuss our implementation of the scheme and some empirical evaluation of it. In Section 8 we discuss limitations of our scheme, and in particular robustness to editing. We follow with open problems and conclusions.

## 2 Model and Preliminaries

Many of the notions in this section are adapted from (CGZ23), the last subsection contains our new definitions and theorems for Steganography in LLMs.

## 2.1 Preliminaries

Let $\lambda$ be the security parameter, we denote by $\text{negl}(\lambda)$ any function that is in $O\left(\frac{1}{p(\lambda)}\right)$ for every polynomial $p(\cdot)$. As is standard in Cryptography research, we think of $\lambda$ as the "key size", and of running times that are super-polynomial in $\lambda$ as "infeasible". We denote by $\log$ and $\ln$ the logarithm with base two and the natural logarithm, respectively. For a sequence $s = (\ldots, s_i, \ldots)$ we denote by $s[i:j]$ the sub-sequence $(s_i, \ldots, s_j)$. The Hamming distance between two vectors is defined as the number of indices on which they differ, that

is $\Delta(z, z') := |\{i \mid z_i \neq z'_i\}|$. We denote by $x \parallel y$ the concatenation of the vectors $x, y$, and by $\text{len}(v)$ the dimension of the vector $v$.

**Pseudorandom function (PRF).** Let $\mathcal{F} = \{F_k : \{0,1\}^{\ell_1(\lambda)} \to \{0,1\}^{\ell_2(\lambda)} \mid k \in \{0,1\}^\lambda\}$ be a family of functions. $\mathcal{F}$ is a PRF if $F_k$ is efficiently computable and for all probabilistic polynomial-time distinguishers $D$,

$$\left| \Pr_{k \leftarrow \{0,1\}^\lambda} \left[ D^{F_k(\cdot)}(1^\lambda) = 1 \right] - \Pr_f \left[ D^{f(\cdot)}(1^\lambda) = 1 \right] \right|$$
$$\leq \text{negl}(\lambda)$$

where $f$ denotes a random function from $\{0,1\}^{\ell_1(\lambda)}$ to $\{0,1\}^{\ell_2(\lambda)}$. PRFs are a standard cryptographic primitive equivalent to one-way functions and can be constructed from standard assumptions (GGM86; HILL99). Intuitively, a PRF is simply a function that is indistinguishable from a totally random function without knowledge of the secret key $k$ that parameterizes it.

## 2.2 Language Models

We adapt the definition of (CGZ23) for language models, which in turns follows that of (KGW$^+$23b). We will often refer to language models simply as *models*.

**Definition 2.1.** A language model $\mathsf{Model}$ over token set $\mathcal{T}$ is a deterministic algorithm that takes as input a prompt $\textsc{prompt}$ and tokens previously output by the model $x = (x_1, \ldots, x_{i-1})$, and outputs a probability distribution $p_i = \mathsf{Model}(\textsc{prompt}, x)$ over $\mathcal{T}$.

A language model $\mathsf{Model}$ is used to generate text as a response to a prompt by iteratively sampling from the returned distribution until a special terminating token $\mathtt{done} \in \mathcal{T}$ is drawn.

**Definition 2.2.** A language model's *response* to $\textsc{prompt}$ is a random variable $\overline{\mathsf{Model}}(\textsc{prompt}) \in \mathcal{T}^\star$ that is defined algorithmically as follows. We begin with an empty list of tokens $x = ()$. As long as the last token in $x$ is not $\mathtt{done}$, we draw a token $x_i$ from the distribution $\mathsf{Model}(\textsc{prompt}, x)$ and append it to $x$. Finally, we set $\overline{\mathsf{Model}}(\textsc{prompt}) = x$.

We assume that our model never outputs text of length super-polynomial in $\lambda$.

## 2.3 Entropy and Empirical Entropy

For a probability distribution $D$ over elements of a finite set $X$, the Shannon *entropy* of $D$ is

$$H(D) = \mathbb{E}_{x \sim D}[-\log D(x)],$$

where $D(x)$ is the probability of $x$ in the distribution $D$. The *empirical entropy* (also known as Shannon information) of $x$ in $D$ is simply $-\log D(x)$. The expected empirical entropy of $x \sim D$ is $H(D)$.

The following definition of empirical entropy of a model's response is taken from (CGZ23).

**Definition 2.3.** For a language model $\mathsf{Model}$, a prompt $\textsc{prompt}$, and a possible response $x \in \mathcal{T}^\star$, we define the *empirical entropy* of $\overline{\mathsf{Model}}$ responding with $x$ to $\textsc{prompt}$, denoted by $H_e(\mathsf{Model}, \textsc{prompt}, x)$, as

$$-\log \Pr\left[ \overline{\mathsf{Model}}(\textsc{prompt}) = x \right].$$

Note that in expectation, Definition 2.3 simply captures the entropy in the response generation. That is, we have

$$\mathbb{E}_x \left[ H_e(\mathsf{Model}, \textsc{prompt}, x) \right] = H\left( \overline{\mathsf{Model}}(\textsc{prompt}) \right),$$

where $x \sim \overline{\mathsf{Model}}(\textsc{prompt})$.

The following definition naturally generalizes empirical entropy from whole outputs to *substrings* out of a model's output.

**Definition 2.4.** For a language model Model, a prompt PROMPT, a possible response $x \in \mathcal{T}^\star$, and indices $i, j \in [\|x\|]$ with $i \leq j$ we define the *empirical entropy on substring* $[i, j]$ of $\overline{\text{Model}}$ responding with $x$ to PROMPT as

$$
H_e^{[i,j]}(\text{Model}, \text{PROMPT}, x) :=
$$
$$
- \log \Pr \left[ \overline{\text{Model}} \left( \text{PROMPT} \right) [i : j] = x[i : j] \; \mid \right.
$$
$$
\left. \overline{\text{Model}} \left( \text{PROMPT} \right) [1 : (i - 1)] = x[1 : (i - 1)] \right].
$$

### 2.3.1 Empirical Entropy in Natural Language

Studies in linguistics (GC02; CLA17; SL22) conclude that in natural language the entropy per unit of text (e.g., a word) is usually constant throughout the text. In particular, the empirical entropy of a LLM response is expected to be linear in the length of the text, and roughly uniformly distributed among the different tokens that assemble the response. This intuition was empirically verified by (CGZ23). We reaffirm the above in Section 7 in which we also run empirical evaluations.

## 2.4 Steganography for LLMs

In this section we finally define rigorously steganography for language models. We first explain the definition intuitively. During the setup of the scheme, we generate a secret key $k$ of size $\lambda$. To generate a response, we use a method $\text{Steg}_k$ that together with the key $k$, receives a prompt PROMPT and a secret message PAYLOAD. A retrieval method $\text{Retr}_k$ should be able to retrieve the hidden PAYLOAD from an output generated using $\text{Steg}_k$, while also using the secret key $k$.

**Definition 2.5** (Steganography Scheme). A *steganography scheme* for a model Model over $\mathcal{T}$ is a tuple of algorithms $\mathcal{W} = (\text{Setup}, \text{Steg}, \text{Retr})$ where:

- $\text{Setup}(1^\lambda) \to k$ outputs a secret key, with respect to a security parameter $\lambda$.

- $\text{Steg}_k(\text{PROMPT}, \text{PAYLOAD})$ is a randomized algorithm that takes as input a prompt PROMPT and a payload PAYLOAD, and generates a response in $\mathcal{T}^\star$.

- $\text{Retr}_k(x) \to \mathcal{T}^\star$ is an algorithm that takes as input and returns as an output sequences in $\mathcal{T}^\star$.

The most important property of the steganography scheme we present in this paper is *undetectability*. Intuitively, we require that without knowledge of the secret key, $\text{Steg}_k(\text{PROMPT}, \star)$ and $\overline{\text{Model}}(\text{PROMPT})$ are indistinguishable even to a user allowed to make many adaptive queries. The payloads used in $\text{Steg}_k$ do not affect this property.

**Definition 2.6** (Undetectability). A steganography scheme $\mathcal{W} = (\text{Setup}, \text{Steg}, \text{Retr})$ is *undetectable* if for every security parameter $\lambda$ and all polynomial-time distinguishers $D$,

$$
\left| \Pr[D^{\text{Model}, \overline{\text{Model}}}(1^\lambda) \to 1] \; - \right.
$$
$$
\left. \Pr_{k \leftarrow \text{Setup}(1^\lambda)} [D^{\text{Model}, \text{Steg}_k}(1^\lambda) \to 1] \right|
$$
$$
\leq \text{negl}(\lambda),
$$

where the notation $D^{\mathcal{O}_1, \mathcal{O}_2}$ means that $D$ is allowed to adaptively query both $\mathcal{O}_1$ and $\mathcal{O}_2$ with arbitrary prompts.

Another important desired property is that $\text{Retr}_k$ should succeed in retrieving the PAYLOAD from the output of $\text{Steg}_k(\text{PROMPT}, \text{PAYLOAD})$. Such a successful retrieval inherently requires making assumptions on the length of the payload and the entropy of the model. It is for example impossible to encode an arbitrary payload

longer than the output in the output, or to plant any payload at all in a deterministic model (thus, without any entropy) while remaining undetectable. The best possible result is thus encoding $H_e(\mathsf{Model}, \textsc{prompt}, x)$ bits of payload in an output $x \leftarrow \mathsf{Steg}_k(\textsc{prompt}, \textsc{payload})$, this is because the empirical entropy of an output exactly quantifies the number of entropy bits used for its generation.

We achieve the above best-possible bound asymptotically yet we require two additional technical conditions. First, we can plant payloads only when $H_e(\mathsf{Model}, \textsc{prompt}, x) = \Omega(\lambda)$. Intuitively, this is because outputs with a very low empirical entropy (with respect to the security parameter) show up too frequently and thus can't be modified while maintaining undetectability. The necessity of this condition, that also appeared in the CGZ result, is proven in (CGZ23). Second, we require that the empirical entropy is spread somewhat uniformly throughout the output $x$. This appears to be necessary for technical reasons (e.g., avoiding a scenario in which the empirical entropy is high solely due to a single very-low-probability token in the response), yet could potentially be relaxed in future works. As discussed in Section 2.3.1 and empirically reaffirmed in Section 7, this condition is satisfied by natural language in which the entropy is roughly constant per each unit of text.

A semi-formal version of the theorem follows, the formal one appears in Section 6.

**Theorem** (Informal version of Theorem 6.3)**.** *Fix a model* $\mathsf{Model}$*. Let* $\textsc{prompt}, \textsc{payload}$ *be strings. Conditioned on the empirical entropy of a response* $y$ *generated by* $\mathsf{Steg}_k(\textsc{prompt}, \textsc{payload})$ *being* **high enough***, the expected length of the prefixes of* $\textsc{payload}$ *and* $\mathsf{Retr}_k(y)$ *that identify is at least* $\Theta(len(y))$*.*

The definition of **high enough**, formally stated in Section 6, roughly means that for any consecutive part of the response consisting of a large enough number $r$ of tokens, the empirical entropy in that part is at least the square root $\tilde{\Omega}(\sqrt{r})$ of the length. As discussed in Section 2.3.1 and verified empirically in Section 7, in natural language we actually expect the entropy to grow *linearly* with the length of the text, much higher than the required square root. Under this condition, the theorem guarantees that a response of length $L$ will allow retrieving the first $\Theta(L)$ bits of the $\textsc{payload}$, which is (up to constants) the best possible.

# 3 Overview of the CGZ Watermark

This section is adapted in its entirety from CGZ (CGZ23), and contains a high-level description of their watermarking scheme.

We first simplify the definition of a language model (Definition 2.1) by assuming that the token set is binary, $\mathcal{T} = \{0, 1\}$. We may assume this without loss of generality due to a straightforward reduction that appears in Section 4.1 of CGZ. We will implicitly use this reduction throughout our work as well.

The intuitive idea behind the CGZ watermark is planting a watermark not by changing the model's distribution, but by correlating the randomness used by the model with the secret key. We begin by describing a simplified approach that works only for generating a single response, of length bounded by some parameter $L$. Let $k = (k_1, k_2, \ldots, k_L)$ be the secret key, chosen by drawing each $k_i$ uniformly and independently from $[0, 1]$. To generate a response to a prompt $\textsc{prompt}$, we run the model as intended yet use $k_i$ to determine the random choice in the $i$-th response token generation. Let $p_i$ denote the probability, according to the real model with the previously chosen tokens as prefix, of the $i$-th token being 1. The watermarked model outputs $x_i = 1$ if $k_i \leq p_i$ and $x_i = 0$ otherwise. Crucially, as $k_i$ was chosen uniformly, the probability of $x_i = 1$ is exactly $p_i$ and hence the output distribution of the model is not affected at all. On the other hand, we now expect some correlation between $x_i$ and $k_i$.

For each response bit $x_i$, the detection algorithm may compute the following score, that depends on the key and on the response but not on the prompt, the model, or the distributions $p_i$,

$$s(x_i, k_i) = \begin{cases} \ln \frac{1}{k_i} & \text{if } x_i = 1 \\ \ln \frac{1}{1-k_i} & \text{if } x_i = 0 \end{cases}.$$

Given a string $x = (x_1, \ldots, x_\ell)$, the detection algorithm sums the score of all bits

$$c(x) = \sum_{i=1}^{\ell} s(x_i, k_i).$$

The main observation is that the score is higher in responses generated by the above procedure than it is for unrelated strings. In non-watermarked text, the value of $k_i$ is independent of the value of $x_i$. Therefore, $s(x_i, k_i)$ is simply an exponential random variable with mean 1:

$$\mathbb{E}_{k_i}[s(x_i, k_i)] = \int_0^1 \ln(1/x) \; dx = 1,$$

and we have $\mathbb{E}_k[c(x) - |x|] = 0$. For watermarked responses, on the other hand,

$$\begin{aligned}
\mathbb{E}_{k_i}[s(x_i, k_i)] &= \int_0^{p_i} \ln \frac{1}{x} \; dx + \int_{p_i}^1 \ln \frac{1}{1-x} \; dx \\
&= \int_0^{p_i} \ln \frac{1}{x} \; dx + \int_0^{1-p_i} \ln \frac{1}{x} \; dx \\
&= 1 + -p_i \cdot \ln p_i - (1 - p_i) \cdot \ln(1 - p_i) \\
&= 1 + \ln(2) \cdot H(p_i),
\end{aligned}$$

and the total expected score is

$$\mathbb{E}_k[c(x) - |x|] = \ln 2 \cdot H(\overline{\mathsf{Model}}(\textsc{prompt})).$$

We thus observed that at least in expectation, the score of watermarked texts is larger than that of arbitrary texts, and that the difference between those quantities is roughly the entropy in the response. To make this observation algorithmically useful, considering expectations is not sufficient, as we need to set some score threshold for detection and analyze the probability of the score passing this threshold in each of the two cases. This analysis will not be repeated in this short overview and appears in CGZ.

To avoid having an upper bound $L$ on the length of the response, and to reduce the length of the key $k$, we use a Pseudo-Random Function $F$ (PRF, as defined in Section 2). The key will now simply be a random string of length $\lambda$, and we would implicitly set $k_i := F_k(i)$. By the definition of a PRF, those values are indistinguishable from independently chosen random values.

The final obstacle is remaining undetectable even when many queries are allowed. In the above sketch the choice of the key $k$ fully determines all randomness, and thus for example the same prompt will always get the same response. To overcome this hurdle, we begin the generation of each response with using real randomness (and not the key) to sample tokens, while counting the empirical entropy of the response prefix generated so far. When the response prefix passes a threshold of empirical entropy $\lambda$, we denote the response's prefix as $r$ and start running the previous scheme with $r$ as an additional input to the PRF. That is, after we set the prefix $r$ we use the value $F_k(r, i)$ to generate the $i$-th token. In the detection, we will enumerate over all possible prefixes of the response as $r$. In CGZ, it is shown that because the prefix $r$ is set only after enough entropy was used, it has negligible probability to ever repeat itself in different queries. Thus the inputs to the PRF calls are each unique and the scheme becomes undetectable even with many queries being made.

The pseudo-codes for generation (Algorithm 1) and detection (Algorithm 2) of the watermark appear in the Appendix. In CGZ, those algorithms are then generalized to also support the detection of the watermark from a substring out of the response and not only from the response in its entirety as is sketched above.

## 4 High-Level Overview of Our Scheme

In this section we give an overview of our construction, with the rigorous details appearing in Sections 5 and 6. As in the CGZ overview in Section 3, we again assume without loss of generality that the token space is binary.

---

**Algorithm 1:** Watermarking algorithm $\mathsf{Wat}_k$

---

**Data:** A prompt (PROMPT) and a secret key $k$
**Result:** Watermarked text $x_1, \ldots, x_L$
$i \leftarrow 1$;
$H \leftarrow 0$;
**while** *done* $\notin (x_1, \ldots, x_{i-1})$ **do**
    $p_i \leftarrow \mathsf{Model}(\text{PROMPT}, x_1, \ldots, x_{i-1})$;
    **if** $H < \lambda$ **then**
        // Collect more internal entropy
        Sample $(x_i, p) \leftarrow (1, p_i)$ with probability $p_i$, otherwise $(0, 1 - p_i)$;
        $H \leftarrow H - \log p$;
        **if** $H \geq \lambda$ **then**
            $r \leftarrow (x_1, \ldots, x_i)$;
        **end**
    **else**
        // Embed the watermark
        $x_i \leftarrow 1[F_k(r, i) \leq p_i]$;
    **end**
    $i \leftarrow i + 1$;
**end**

---

**Algorithm 2:** Detector $\mathsf{Detect}_k$

---

**Data:** Text $x_1, \ldots, x_L$ and a secret key $k$
**Result:** True or False
**for** $i \in [L]$ **do**
    $r^{(i)} \leftarrow (x_1, \ldots, x_i)$;
    Define $v_j^{(i)} := x_j \cdot F_k(r^{(i)}, j) + (1 - x_j) \cdot (1 - F_k(r^{(i)}, j))$ for $j \in [L]$;
    **if** $\sum_{j=i+1}^{L} \ln(1/v_j^{(i)}) > (L - i) + \lambda\sqrt{L - i}$ **then**
        **return** True;
    **end**
**end**
**return** False;

---

As a first attempt, we notice that one may generalize any watermark into a steganography scheme by using several keys. Let $k_1, \ldots, k_m$ be $m$ different secret keys, and setup a watermarking scheme with each of them. To encode a message $i \in [m]$ within a response, simply use the watermarking instance corresponding to $k_i$ to generate said response. In the retrieval step, we will use the detection algorithm with *every* key $k_j$ to find which of them was used. While undetectability is trivially preserved, as we only use undetectable watermarks to generate responses, the scheme becomes infeasible as soon as $m$ isn't very small. This is because both the rate of "false-positives" and the detection time grow by a multiplicative factor of $m$. In particular, encoding $\ell$ bits of information will cause a multiplicative factor of $2^\ell$ in the running time of the retrieval algorithm, and will also require that the false-positive rate of the watermarking scheme be much smaller than $2^{-\ell}$.

A reasonable next step then, is breaking up the payload into smaller parts (say, characters or bits), and encoding each of those parts separately in a similar fashion to the previous suggestion. One subtle issue to overcome while implementing this idea is that partitioning the response into those smaller chunks is not straightforward. This is because we know a successful watermarking requires high empirical entropy, and it is not known in advance what parts of the response would contain high empirical entropy. Moreover, the retriever needs to be able to use the same partition as the steganography algorithm. We solve this problem by implicitly defining the partition to chunks using the detection score itself: Let $t$ be some score threshold to be decided later. Denote the first bit of the payload by $b \in \{0, 1\}$. We start planting the payload in the same way as the CGZ watermark is embedded, but with $b$ as an additional input to the PRF. That is,

the randomness used in the $i$-th token is $F_k(i, b)$. The retriever would compute the scores for both options of $b$ over more and more tokens until one of the two scores passes a certain threshold determined by $t$. For example, we can consider the minimal prefix of the response $x[: i]$ for which $(c(x[: i]) - i)/\sqrt{i} > t$ with respect to the score computed with either $b = 0$ or $b = 1$. Crucially, *both* $\mathsf{Steg}_k$ and $\mathsf{Retr}_k$ can compute those scores, so both can compute the index $i$ in which one of the scores passed the threshold for the first time. Thus, both are able to agree on the end of the chunk encoding the first bit of the payload, and thus can iteratively restart this process with the next bit of the payload and so on. Following the CGZ analysis, a threshold of roughly $t \approx \lambda$ is required to make sure the correct bit is always the first one to pass the score threshold, as otherwise the false-positive rate of passing the score threshold is non-negligible. This means that whilst the decoding time and general error rate are now low enough, we still can't encode payloads that are too long. Even if every token contains constant entropy, only $\Theta(L/\lambda)$ bits of the payload can be encoded in a response of length $L$. Since $\lambda$ is rather large, we should desire to get this quantity down to the optimal $\Theta(L)$.

Finally, we use Error Correcting Codes (ECC) to lower the threshold $t$ and in turn increase the number of hidden payload bits. Intuitively, if instead of setting $t \approx \lambda$ we set $t$ to be a large constant, then false-positives do happen, but only with a small constant probability. Thus, if instead of encoding the bits of the payload itself, we would encode the bits of an ECC applied to the payload, we should be able to recover the payload even if a constant fraction of the bits were transmitted incorrectly. This would reduce $t$ from $\lambda$ to $O(1)$ as desired. A problem that remains is that applying a standard ECC "scrambles" the message and thus if only a prefix of the code is successfully hidden then it would be impossible to decode anything out of the payload. For this reason, in Section 5 we define and construct a *Dynamic* ECC, this is a code in which for every $i$, the first $i$ bits of the code can be used to retrieve the first $\Theta(i)$ bits of the message. Thus, a response of size $L$ would hide the first $\Theta(L)$ bits of the payload. To significantly simplify the construction of the ECC, we use the fact our construction provides *noiseless feedback*. As mentioned before, during the encoding process, $\mathsf{Steg}_k$ can also simulate $\mathsf{Retr}_k$ and thus knows what bit will be detected by the retriever at each step. Thus, the ECC can depend on whether or not each previously sent bit was transmitted correctly. In the construction of Section 5 we actually use a ternary code alphabet rather than binary, which doesn't affect the sketch of the construction much.

We finally note that to support multiple queries, we use the same idea of CGZ sketched in Section 3, and begin by observing enough real entropy to set a unique prefix $r$, the following random choices will be made using $F_k(r, i, b)$. For the detector to find the correct prefix to use as $r$, we need to use the first $\lambda$ bits of entropy after setting the prefix to place a normal watermark, which the detector would use to verify the choice of $r$. This means that to be undetectable with many queries, we start encoding the payload only after the first $\Theta(\lambda)$ tokens (or bits of entropy). As long as $L = \Omega(\lambda)$ this does not matter asymptotically.

## 5 Dynamic Error Correcting Code

Error Correcting Codes (ECCs) are the means to compensate for errors in the transmission of messages. An ECC encoding is a function $\mathrm{Enc} : \Sigma^k \to \Gamma^n$ from messages of length $k$ over alphabet $\Sigma$, to codewords of length $n$ over alphabet $\Gamma$. The *rate* of a code is $R(\mathrm{Enc}) := \frac{k}{n}$, which signifies how efficient the code is. The *(relative) distance* of a code is $\delta(\mathrm{Enc}) := \frac{1}{n} \min_{z \neq z'} \mathrm{Enc}(z) \Delta \mathrm{Enc}(z')$, which is twice the fraction of corrupt indices in a codeword that still allows decoding it to the original message. A code (or a family of codes) is considered *good* if both its rate and distance are constant, which means that the length of messages is only expanded by a constant factor, yet a constant fraction of errors can be corrected. ECCs are extensively studied and it is long known that good ECCs can be constructed, even when $\Sigma = \Gamma = \mathbb{F}_2$. (Ham50; Gil52; Var57; Jus72; SS96)

An ECC with *feedback* is an ECC in which we transmit the symbols of the codeword $\mathrm{Enc}(x)$ one-by-one, and immediately receive feedback with regards to whether an error occurred in transmitting this symbol. The following symbols we submit may adaptively depend on the feedback received so far. We say that the feedback is *noiseless* if the feedback received is always reliable. If the errors in transmission occur randomly (i.e., each transmitted symbol has the same probability of becoming corrupted), then it turns out that noiseless feedback does not improve the parameters of the best possible ECC. On the other hand, if the small fraction of corrupted symbols is chosen adversarially, then noiseless feedback does improve the best possible distance. Feedback also appears to allow simpler and more efficient encoding and decoding schemes. (Ber64; Cov88)

We define a natural generalization of ECCs, in which the length of the message (and hence also of the code) is not known in advance. We call those *Dynamic* ECCs. We would require that for *any* $k' \leq k$, the first $k'$ symbols of the message can be decoded from the first $O(k')$ symbols of the codeword, even if a small fraction of those codeword symbols are corrupted. This definition is similar yet weaker than the definition of Tree Codes (Sch93; Sch96).

**Definition 5.1.** For alphabets $\Sigma, \Gamma$ a family $\{\mathrm{Enc}_k\}_{k \in \mathbb{N}}$ of functions $\mathrm{Enc}_k : \Sigma^k \to \Gamma^\star$ is called a Dynamic ECC if for every $k \in \mathbb{N}$, the function $\overline{\mathrm{Enc}_k} : \Sigma^k \to \Gamma^{n_k}$ is an ECC, where

$$\overline{\mathrm{Enc}_k}(x) := \mathrm{Enc}_1(x[:1]) \;||\; \mathrm{Enc}_2(x[:2]) \;||\; \ldots \;||\; \mathrm{Enc}_k(x),$$

$$n_k := \max_{x \in \Sigma^k} \mathrm{len}(\overline{\mathrm{Enc}_k}(x)).$$

In simple words, a Dynamic ECC is a family of standard ECCs where the codeword corresponding to the a prefix of a message, is always a prefix of the codeword corresponding to the entire message.

**Definition 5.2.** The rate of a Dynamic ECC is $R(\mathrm{Enc}) := \inf_{k \in \mathbb{N}} R(\overline{\mathrm{Enc}_k}) = \inf_{k \in \mathbb{N}} \frac{k}{n_k}$. The distance of it is $\delta(\mathrm{Enc}) := \inf_{k \in \mathbb{N}} \delta(\overline{\mathrm{Enc}_k})$.

In a similar manner, we also define a Dynamic ECC with (noiseless) feedback to be a Dynamic ECC in which after each symbol transmitted we receive a feedback as to which symbol was received. We next present a simple construction of a Dynamic ECC with feedback where $|\Sigma| = 2, |\Gamma| = 3$, and both the rate and distance are constant. This construction is rather straightforward and is similar to constructions used in slightly different settings (EGH15).

**Theorem 5.3.** *For any $\varepsilon \in (0, \frac{1}{2})$ there exists a Dynamic ECC with noiseless feedback with $|\Sigma| = 2, |\Gamma| = 3$, in which $\varepsilon$ fraction of errors can be corrected and $n_k = \lceil \frac{k}{1-2\varepsilon} \rceil$. Both encoding and correction take linear time.*

We think of the message alphabet as binary $\Sigma = \{0, 1\}$, and to the codeword alphabet we add an additional symbol $\Gamma = \{0, 1, \leftarrow\}$. We would think of the symbol '$\leftarrow$' as a "backspace". Intuitively, we will always compute the message that is the decoding of what the receiver saw so far, and if it is consistent with the input we simply send the next bit of the input. If it is not consistent with input, we will send a "backspace" to indicate that the last symbol is incorrect. We will do so iteratively.

For a sequence $y = (y_1, \ldots, y_n) \in \Gamma^\star$, we recursively define $\mathrm{decode}(y)$ to be $\mathrm{decode}(y[:(n-1)]) \;||\; (y_n)$ if $y_n \in \{0, 1\}$, and $\mathrm{decode}(y[:(n-1)])[:-1]$ if $y_n =' \leftarrow'$, where $v[:-1]$ means removing the last symbol from the vector $v$ (unless its empty). As the base case, we have $\mathrm{decode}(()) = ()$.

For a message $x = (x_1, \ldots, x_k) \in \Sigma^\star$ and a previously transmitted (partial) codeword $y = (y_1, \ldots, y_n) \in \Gamma^\star$ we define the longest agreeing prefix of $x$ and the decoding of $y$ as

$$\mathrm{last}(x, y) := \max_i \{i \mid x[:i] = \mathrm{decode}(y)[:i]\}.$$

We then define the length of the wrong suffix of the decoding of $y$ as $\mathrm{suff}(x, y) := \mathrm{len}(\mathrm{decode}(y)) - \mathrm{last}(x, y)$.

Given a message $x$ and partial codeword $y$, we define the next symbol to be sent as $\mathrm{next}(x, y) =' \leftarrow'$ if $\mathrm{suff}(x, y) > 0$, and as $\mathrm{next}(x, y) = x[\mathrm{last}(x, y) + 1]$ otherwise. Our protocol is thus simple, if $x$ is the message and $y$ is the codeword received by the receiver so far (which we know using the noiseless feedback), then the next symbol we send is $\mathrm{next}(x, y)$.

**Lemma 5.4.** *Let $x \in \Sigma^\star$ be a message and $y \in \Gamma^n$ be a partial codeword received by the receiver according to the described protocol, and assume that at most $\varepsilon n$ of the symbols in $y$ were received differently than sent by the protocol. Then, $\mathrm{last}(x, y) \geq (1 - 2\varepsilon)n$.*

*Proof.* For any partial received codeword $y'$ we define the potential function $\Phi(x, y') := \mathrm{last}(x, y') - \mathrm{suff}(x, y')$.

We first show that if the next token is received correctly then the potential increases by one, that is, $\Phi(x, y' \;||\; \mathrm{next}(x, y')) = \Phi(x, y') + 1$. We show this by considering two cases. If $\mathrm{suff}(x, y') = 0$ then $\mathrm{decode}(y') = x[:\mathrm{last}(x, y')]$ and $\mathrm{next}(x, y') = x[\mathrm{last}(x, y') + 1]$, thus $\mathrm{decode}(x, y' \;||\; \mathrm{next}(x, y')) =$

$x[\colon \mathrm{last}(x, y') + 1]$. Otherwise, $\mathrm{suff}(x, y') > 0$ and $\mathrm{next}(x, y') =' \leftarrow'$, and hence $\mathrm{suff}(x, y' \mid\mid \mathrm{next}(x, y')) = \mathrm{suff}(x, y') - 1$.

Next, we show that if the next token is received incorrectly then the potential decreases by one, that is $\Phi(x, y' \mid\mid s) = \max(0, \Phi(x, y') - 1)$ whenever $s \neq \mathrm{next}(x, y')$. We again consider two cases. If $\mathrm{suff}(x, y') > 0$ then we have $s \in \{0, 1\}$ and in turn $\mathrm{suff}(x, y' \mid\mid s) = \mathrm{suff}(x, y') + 1$. Otherwise $\mathrm{suff}(x, y') = 0$ and we either have $\mathrm{suff}(x, y' \mid\mid s) = 1$ if $s \neq' \leftarrow'$ or have $\mathrm{last}(x, y' \mid\mid s) = \max(0, \mathrm{last}(x, y') - 1)$ if $s =' \leftarrow'$.

We conclude that if $e$ out of the $n$ symbols in $y$ were received incorrectly, then

$$\Phi(x, y) \geq 1 \cdot (n - e) - 1 \cdot e = n - 2e \geq n - 2 \cdot \varepsilon n.$$

On the other hand, as $\mathrm{suff}(x, y) \geq 0$ we also have $\mathrm{last}(x, y') \geq \Phi(x, y')$. □

*Proof of Theorem 5.3.* Denote by $n_k = \lceil \frac{k}{1 - 2\varepsilon} \rceil$. Let $x$ be a message and $y$ the first $n_k$ tokens received by running the protocol. Assume that at most $\varepsilon n_k$ out of those tokens were received incorrectly. By Lemma 5.4, we have

$$\mathrm{last}(x, y) \geq (1 - 2\varepsilon) n_k \geq k.$$

Hence, $\mathrm{decode}(y)[\colon k]$ correctly retrieves $x[\colon k]$. □

## 6 Our Scheme

As in the overview of Section 3, we begin by analysing a scheme in which only a single query is undetectable. Then, in Section 6.1 we apply the same idea of CGZ to go from undetectability for one query to complete undetectability. An intuitive explanation of our scheme is covered in Section 4.

---

**Algorithm 3:** One-query steganography algorithm $\mathsf{Steg}_k$

---

**Data:** A prompt (PROMPT), a payload (PAYLOAD), and a secret key $k$
**Result:** Response $x_1, \ldots, x_L$
$i \leftarrow 1$;
$\mathrm{code} \leftarrow ()$;
$\mathrm{score}_\sigma \leftarrow 0$ for $\sigma \in \{0, 1, \leftarrow\}$;
$\mathrm{score\_len} \leftarrow 0$;
$\mathrm{next} \leftarrow \mathrm{next}(\mathrm{PAYLOAD}, \mathrm{code})$;
**while** *done* $\notin (x_1, \ldots, x_{i-1})$ **do**
    $p_i \leftarrow \mathsf{Model}(\mathrm{PROMPT}, x_1, \ldots, x_{i-1})$;
    $x_i \leftarrow 1[F_k(i, \mathrm{next}) \leq p_i]$;
    $\mathrm{score\_len} \leftarrow \mathrm{score\_len} + 1$;
    **for** $\sigma \in \{0, 1, \leftarrow\}$ **do**
        $\mathrm{score}_\sigma \leftarrow \mathrm{score}_\sigma + s(x_i, F_k(i, \sigma))$;
        **if** $(\mathrm{score}_\sigma - \mathrm{score\_len})/\sqrt{\mathrm{score\_len}} > t$ **then**
            $\mathrm{code} \leftarrow \mathrm{code} \mid\mid (\sigma)$;
            $\mathrm{score}_\sigma \leftarrow 0$ for $\sigma \in \{0, 1, \leftarrow\}$;
            $\mathrm{score\_len} \leftarrow 0$;
            $\mathrm{next} \leftarrow \mathrm{next}(\mathrm{PAYLOAD}, \mathrm{code})$;
            **break**;
        **end**
    **end**
    $i \leftarrow i + 1$;
**end**

---

Algorithm 3 naturally follows the sketch of Section 4, while using the ECC (and notation) of Section 5. Let's informally describe the algorithm in words. The algorithm depends on a score threshold $t$ to be set

later. We apply the Dynamic ECC to the message PAYLOAD, and denote by *next* the next symbol we are supposed to transmit according to the ECC. We start generating response tokens following the CGZ scheme, by comparing the actual distribution $p_i$ to the PRF value $F_k(i, \text{next})$. As the PRF value depends on *next*, a correlation between the randomness we use and the token we are attempting to transmit is created. After each generated response token, we also emulate the retriever: Ignoring our knowledge of *next*, for every code symbol $\sigma \in \{0, 1, \leftarrow\}$ we compute the score (as defined in Section 3) of the response tokens we saw so far. We wait until the score of a some symbol $\sigma$, normalized with respect to the number of tokens involved in the score computation, passes a threshold $t$. That is, until

$$\frac{\text{score}_\sigma - \text{score\_len}}{\sqrt{\text{score\_len}}} > t.$$

When this happens, we view $\sigma$ as the symbol received by the ECC receiver. While $\sigma = \text{next}$ is supposed to be more likely, the symbol could be incorrect. Whenever we add a symbol to the code, we restart our computation of the score and start transmitting the next code symbol. Algorithm 4 shows the retrieval process, which is identical to what is emulated within the steganography algorithm. Note that both algorithms have a linear running time.

---

**Algorithm 4:** One-query retriever $\text{Retr}_k$

**Data:** Response $x_1, \ldots, x_L$, and a secret key $k$
**Result:** Retrieved payload PAYLOAD
code $\leftarrow$ ();
$\text{score}_\sigma \leftarrow 0$ for $\sigma \in \{0, 1, \leftarrow\}$;
$\text{score\_len} \leftarrow 0$;
**for** $i = 1, 2, \ldots, L$ **do**
    $\text{score\_len} \leftarrow \text{score\_len} + 1$;
    **for** $\sigma \in \{0, 1, \leftarrow\}$ **do**
        $\text{score}_\sigma \leftarrow \text{score}_\sigma + s(x_i, F_k(i, \sigma))$;
        **if** $(score_\sigma - score\_len)/\sqrt{score\_len} > t$ **then**
            code $\leftarrow$ code $||$ $(\sigma)$;
            $\text{score}_\sigma \leftarrow 0$ for $\sigma \in \{0, 1, \leftarrow\}$;
            $\text{score\_len} \leftarrow 0$;
            **break**;
        **end**
    **end**
**end**
**return** decode(code);

---

We first observe that the distribution of the response is indistinguishable from the distribution of the original model (when a single query is made).

**Lemma 6.1.** *For any* Model *and any* PROMPT, PAYLOAD, $t$, *the distribution of* $\text{Steg}_k(\textit{PROMPT}, \textit{PAYLOAD})$ *over a random choice of key $k$ is indistinguishable from the distribution of* $\overline{\text{Model}}(\textit{PROMPT})$.

*Proof.* The proof is rather straightforward and follows CGZ and its overview in Section 3. If we replace each time $F_k(i, \text{next})$ is used (to determine $x_i$) with a new uniformly chosen value in $[0, 1]$, then the distribution of $\overline{\text{Model}}$ is completely unaffected. Using a PRF instead of "fresh" random values is indisitinguishable as long as we don't use the PRF on the same input more than once. As each input to the PRF consists of the (unique) index $i$, we never call it on the same input. $\square$

We should next show that $\text{Retr}_k(\text{Steg}_k(\textit{PROMPT}, \textit{PAYLOAD}))$ successfully retrieves PAYLOAD. As discussed in Section 2.4, doing so requires making assumptions on the empirical entropy of the generated response. We prove that a relatively weak assumption (which in particular covers the case of natural languages) is sufficient,

yet it is very likely that the proof can be adapted for other conditions as well - as the algorithm itself is quite flexible. We also note that in the proof we don't optimize for constants but for simplicity (of proof), the empirical evaluation in Section 7 implies that the actual constants are far better than in the following proof.

**Definition 6.2.** Let $h = (h_1, \ldots, h_L)$ be a sequence of empirical entropies (i.e., non-negative numbers). We say that $h$ is $r_0$-saturated if for every consecutive subsequence of $h$ of length $r \geq r_0$, the sum of entropies is at least $10\sqrt{r}\ln r$. That is, for every $r \geq r_0$ and $1 \leq i \leq L - (r - 1)$, we have $\sum_{j=i}^{i+r-1} h_j \geq 10\sqrt{r}\ln r$.

For example, if the empirical entropy in each consecutive block of $b$ tokens is at least some constant $\alpha > 0$, then the empirical entropies are $\tilde{O}\left(\frac{b^2}{\alpha^2}\right)$-saturated. This is because a consecutive block of $bk$ tokens contains at least $\alpha k$ entropy, which is larger than $10\sqrt{bk}\ln(bk)$ if $k = \tilde{\Omega}\left(\frac{b}{\alpha^2}\right)$. Hence, natural language which has this property (as discussed in Section 2.3.1) is $O(1)$-saturated. In fact, the entropy of natural language grows linearly with the length of the text, while our condition is merely for it to grow faster than the square root of the length of the text. We verify these claims empirically in Section 7.

Finally, we prove that if the empirical entropy of a response is $O(1)$-saturated, and the response is of length $L$, then in expectation at least the first $\Theta(L)$ bits of the PAYLOAD are retrieved correctly.

**Theorem 6.3.** *Fix a model* Model *and an integer* $r_0$, *there exists a choice of threshold $t$ for which the following holds. Let* PROMPT, PAYLOAD *be strings. Conditioned on the empirical entropy of a response $y$ generated by* $\mathsf{Steg}_k(\textit{PROMPT}, \textit{PAYLOAD})$ *being $r_0$-saturated, the expected length of the prefixes of* PAYLOAD *and* $\mathsf{Retr}_k(y)$ *that identify is at least* $\Theta(len(y)/r_0)$.

The proof of Theorem 6.3 is deferred to Appendix A.

### 6.1 Complete Undetectability

To move from undetectability of a single response to the general undetectability defined in Definition 2.6, we simply repeat the "trick" of CGZ as overviewed in Section 3. The complete details are deferred to Appendix B.

## 7 Empirical Evaluation

We implemented[1] Algorithms 3 and 4 from Section 6, that provide undetectability for a single query. We did so for simplicity and as we only aim to evaluate the new contributions of this paper.

In Figure 2, we estimate the number of message bits we can hide in a response of a certain length. For each length of response, we ran our scheme for 100 times using the LLM model GPT-2 (RWC[+]19) on a randomly chosen prompt from the list of example prompts provided by OpenAI on their GPT-2 webpage.[2] We ran it with threshold parameter $t = 2$, which we didn't optimize. As expected, it is evident that the number of hidden bits grows linearly with the length of the response. In Figure 3, we detail the decoding process of the example shown in Figure 1, generated using the 7B parameters Instruct version of the Llama 2 model developed by Meta (TMS[+]23). We did not optimize for constants. The takeaways from this section are twofold; First, we show that the implementation of our scheme is simple and the run-time overhead is minimal; Second, we empirically support the claim from Section 2.3.1 that the empirical entropy in LLMs grows linearly with the length of the response.

## 8 Limitations and Open Problems

The main issue we did not discuss so far is robustness to editing. That is, can the payload be recovered even if the model's response is somehow edited? We mainly leave dealing with robustness to future work, yet next list a couple of observations regarding robustness. In CGZ (CGZ23), the watermarking scheme is adapted to "restart" once-in-a-while so that the watermark will be detectable from any long enough consecutive substring of the response (and not only from the entire response). The same modification can easily be applied to

---

[1]Code available at: https://github.com/OrZamir/steg
[2]https://openai.com/research/better-language-models

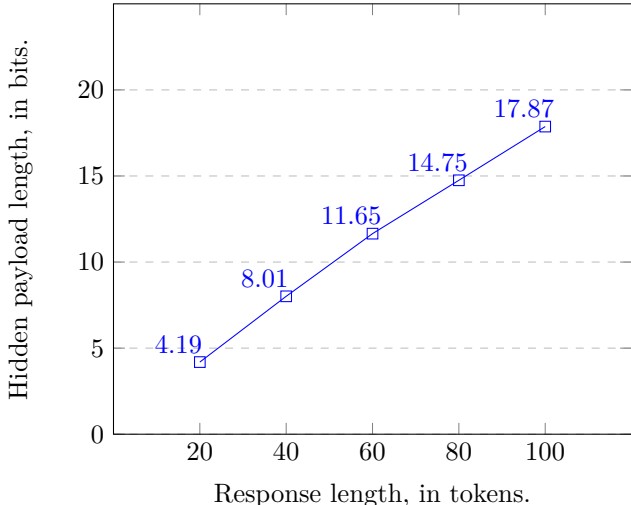

Figure 2: Plot of the number of successfully hidden payload bits, by length of response. Experiments ran on GPT-2 with a random choice of an example prompt taken from the OpenAI website. The experiment was performed 100 times for each response length.

| **Prompt:** | Write an email asking my professor Prof. Hannity to not make the final exam in Machine Learning 101 too difficult. |
|---|---|
| **Response:** | Dear Professor Hannity, I hope this email finds you well. I am writing to respectfully request that you consider the level of difficulty for the final exam in Machine Learning 101. While I am confident in my understanding of the course materials and have put in a significant 0 amount of effort throughout the semester, I do 1 have 1 concerns 0 about the potential 1 difficulty ← of the final exam. As you may recall, several students in my previous sections of ← Machine 1 Learning have found the final exam to be very challenging 1 , leading to frustration 1 and ← disappointment 0 . While I ← understand that the course is intended to push students to their limits and beyond, I believe that some relaxation of the latter 1 may be in order. I would kindly ask that you consider reducing the difficulty of the final exam or offering some additional supports or 1 resources to help students prepare. I 0 believe that this could 1 enhance 0 |
| **Error Correcting Code:** | 01101←←111←0←11010 |
| **Hidden Payload:** | "OZ", encoded as 01111 11010. |

Figure 3: A breakdown of the decoding algorithm for the example in Figure 1.

our scheme as well, making the payload retrievable from any long enough substring out of the model's response. At the other end of the spectrum, it is known that under certain conditions powerful users can edit any watermark out of a model's response (ZEF$^+$23; CGZ23). Intuitively, a complete rephrasing of the response, for example, is supposed to remove any watermark. The previous empirical works on watermarks and steganography, that do not guarantee undetectability, showcase some robustness to certain types of edits.

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

## Appendix

## A    Proof of Theorem 6.3

We need to analyze two quantities. First, when $\mathsf{Steg}_k$ adds a symbol to the *code*, what is the probability it is incorrect (i.e., different than the intended *next* symbol)?; Second, how many symbols do we manage to add to the *code*? Or equivalently, how many response tokens do we usually see before adding a symbol to the *code*?

To answer both questions, we analyze the evolution of the correct score (i.e., the one corresponding to the *next* symbol) and incorrect scores from the time we start computing them and until one of them passes the threshold.

While computing the score with respect to an incorrect symbol, every token's score is simply an exponential random variable with mean 1. Denote by $s_1, s_2, \dots$ the scores of each individual token (i.e., independent $\mathrm{Exp}(1)$ random variables), and by $S_i := \sum_{j=1}^{i} s_j$ their accumulative sums. By Lemma 5 in CGZ we have that for any $\ell, \tau > 0$,

$$\Pr[S_\ell \geq \ell + \tau\sqrt{\ell}] \leq \left(\frac{4}{5}\right)^\tau .$$

Let $b \geq r_0$ be an integer to be chosen later. By a union bound, the probability of the score passing the threshold $t$ at any time within the $b$ first steps is bounded by $b\left(\frac{4}{5}\right)^t$.

For the score with respect to the correct symbol, the first $b$ tokens contain at least $10\sqrt{b}\ln b$ empirical entropy, as $b \geq r_0$ and as we conditioned on our response being $r_0$-saturated. In CGZ it is shown that $S_\ell$ is still distributed as the sum of $\ell$ independent $\mathrm{Exp}(1)$ variables, but it is now additively shifted by the empirical entropy of those $\ell$ variables. In particular, by Theorem 7 and Lemma 5 in CGZ, it follows that for any $\tau > 0$ we have

$$\Pr[S_b < b + 10\sqrt{b}\ln b - \sqrt{\tau b}] \leq e^{-\tau/2}.$$

Equivalently,

$$\Pr\left[\frac{S_b - b}{\sqrt{b}} < 10\ln b - \sqrt{\tau}\right] \leq e^{-\tau/2}.$$

We choose $t = 5\ln b$ and $\tau = (5\ln b)^2$ and deduce from the above statements that: I) The probability that an incorrect (normalized) score passed the threshold $t$ within the first $b$ steps is at most $b\left(\frac{4}{5}\right)^t = b\left(\frac{4}{5}\right)^{5\ln b} = e^{(1-5\ln(5/4))\ln b} < e^{-(\ln b)/10}$. II) The probability that the correct (normalized) score passed the threshold $t$ within the first $b$ steps, which is at least the probability it was above the threshold at the end of the $b$-th step, is at least $1 - e^{-\tau/2} = 1 - e^{-25(\ln b)^2/2}$. By combining (I) and (II) we conclude that the probability that the correct score passed the threshold within the first $b$ steps, yet the two incorrect scores did not, is at least

$$1 - e^{-25(\ln b)^2/2} - 2e^{-(\ln b)/10},$$

denote this number by $(1 - \varepsilon(b))$. As $\lim_{b\to\infty} \varepsilon(b) = 0$, there exists a constant $b_0$ such that for every $b \geq b_0$ it holds that $\varepsilon(b) \leq \frac{1}{3}$. We set $b = \max(r_0, b_0)$. Note that $b_0$ is a universal constant independent of $r_0$ and other parameters. We conclude that with probability at least $\frac{2}{3}$ the correct symbol is transmitted within the first $b$ tokens, and in particular the symbol is transmitted correctly with probability at least $\frac{2}{3}$.

As the probability of incorrectly transmitting a symbol is at most $\frac{1}{3} < \frac{1}{2}$, we can use Theorem 5.3 to conclude that if $n$ code symbols are transmitted overall, then the first $\Theta(n)$ bits of the PAYLOAD are retrieved correctly. It is thus only left to analyze the number of transmitted code symbols.

We again consider the same inequality from before, that holds for any $b' \geq r_0, \tau > 0$, $\Pr\left[\frac{S_{b'} - b'}{\sqrt{b'}} < 10\ln b' - \sqrt{\tau}\right] \leq e^{-\tau/2}$. By choosing $\tau = (5\ln b')^2$ we observe that for any $b' > b$ we have $10\ln b' - \sqrt{\tau} = 5\ln b' > 5\ln b = t$. And thus, $\Pr\left[\frac{S_{b'} - b'}{\sqrt{b'}} < t\right] \leq e^{-25(\ln b')^2/2}$. Denote by $\ell$ the random variable which is the first step in which the score (w.r.t. the correct symbol) passed the threshold $t$,

by the above inequality we have

$$\mathbb{E}\left[\ell\right] = \sum_{i=1}^{\infty} \Pr\left[\ell \geq i\right]$$

$$= \sum_{i=1}^{b} \Pr\left[\ell \geq i\right] + \sum_{i=b+1}^{\infty} \Pr\left[\ell \geq i\right]$$

$$< \sum_{i=1}^{b} 1 + \sum_{i=b+1}^{\infty} e^{-\frac{25}{2}(\ln i)^2}$$

$$= b + O(1).$$

As the correct symbol is expected to pass the score threshold after $b + O(1)$ response tokens, in particular a symbol is expected to be transmitted in the protocol at least once every $b + O(1)$ response tokens.

## B  Complete Undetectability

Our revised algorithm partitions the generation of response tokens into three parts:

1. We use real randomness to generate tokens and count the amount of empirical entropy used in the process, until enough (at least $\lambda$) empirical entropy was seen, we call the prefix of tokens generated in this step $r$.

2. We begin generating tokens using the PRF, with both $r$ and the index as inputs. We don't yet submit code symbols and wait until the score passes $\lambda$. We do this step to leave a signal to the detector with regards to what prefix $r$ was chosen by the generation algorithm.

3. We now run the one-query scheme, while adding $r$ as an additional input to the PRF calls.

We remark that the second part can be avoided, as the correct $r$ can be detected implicitly during the third part, but for simplicity of presentation we include it. With this structure, the proofs are rather straightforward. The pseudo-code for $\mathsf{Steg}_k$ appears in Algorithm 5, and for $\mathsf{Retr}_k$ in Algorithm 6.

**Theorem B.1.** *Algorithm 5 is undetectable as defined in Definition 2.6.*

*Proof.* This follows from Theorem 11 in CGZ, as the inputs to the PRF in the generation process are all unique within a single query, and all contain $r$ which ever repeats within different queries only with negligible probability. $\qquad\square$

Theorem 6.3 also remains correct as-is, besides that we "lose" $O(\lambda)$ empirical entropy to Parts 1 and 2. As long as $L = \Omega(\lambda)$ then, the same theorem statement still holds. While the running time of $\mathsf{Steg}_k$ is unaffected, the running time of $\mathsf{Retr}_k$ is now quadratic instead of linear. This can be avoided by truncating the verification of $r$, but we do not do so for the sake of simplicity.

---

**Algorithm 5:** Steganography algorithm $\mathsf{Steg}_k$

---

**Data:** A prompt (PROMPT), a payload (PAYLOAD), and a secret key $k$
**Result:** Response $x_1, \ldots, x_L$
$i \leftarrow 1$;
$H \leftarrow 0$;
r_score $\leftarrow 0$;
r_score_len $\leftarrow 0$;
code $\leftarrow ()$;
$\text{score}_\sigma \leftarrow 0$ for $\sigma \in \{0, 1, \leftarrow\}$;
score_len $\leftarrow 0$;
next $\leftarrow \text{next}(\text{PAYLOAD}, \text{code})$;
**while** *done* $\notin (x_1, \ldots, x_{i-1})$ **do**
    $p_i \leftarrow \mathsf{Model}(\text{PROMPT}, x_1, \ldots, x_{i-1})$;
    **if** $H < \lambda$ **then**
        // Part 1
        Sample $(x_i, p) \leftarrow (1, p_i)$ with probability $p_i$, otherwise $(0, 1 - p_i)$;
        $H \leftarrow H - \log p$;
        **if** $H \geq \lambda$ **then**
            $r \leftarrow (x_1, \ldots, x_i)$;
        **end**
    **else if** $(r\_score - r\_score\_len) \leq \lambda \sqrt{r\_score\_len}$ **then**
        // Part 2
        $x_i \leftarrow 1[F_k(r, i, \text{None}) \leq p_i]$;
        r_score $\leftarrow$ r_score $+ s(x_i, F_k(r, i, \text{None}))$;
        r_score_len $\leftarrow$ r_score_len $+ 1$;
    **else**
        // Part 3
        $x_i \leftarrow 1[F_k(r, i, \text{next}) \leq p_i]$;
        score_len $\leftarrow$ score_len $+ 1$;
        **for** $\sigma \in \{0, 1, \leftarrow\}$ **do**
            $\text{score}_\sigma \leftarrow \text{score}_\sigma + s(x_i, F_k(r, i, \sigma))$;
            **if** $(score_\sigma - score\_len)/\sqrt{score\_len} > t$ **then**
                code $\leftarrow$ code $|| (\sigma)$;
                $\text{score}_\sigma \leftarrow 0$ for $\sigma \in \{0, 1, \leftarrow\}$;
                score_len $\leftarrow 0$;
                next $\leftarrow \text{next}(\text{PAYLOAD}, \text{code})$;
                **break**;
            **end**
        **end**
    **end**
    $i \leftarrow i + 1$;
**end**

---

---

**Algorithm 6:** Retriever algorithm $\mathsf{Retr}_k$

---

**Data:** Response $x_1, \ldots, x_L$, and a secret key $k$
**Result:** Retrieved payload PAYLOAD
code $\leftarrow ()$;
score$_\sigma \leftarrow 0$ for $\sigma \in \{0, 1, \leftarrow\}$;
score_len $\leftarrow 0$;
**for** $j = 1, 2, \ldots, L$ **do**
  $r \leftarrow (x_1, \ldots, x_j)$;
  r_score $\leftarrow 0$;
  r_score_len $\leftarrow 0$;
  **for** $i = j+1, j+2, \ldots, L$ **do**
    **if** $(r\_score - r\_score\_len) \leq \lambda\sqrt{r\_score\_len}$ **then**
      // Verify $r$
      r_score $\leftarrow$ r_score $+ s(x_i, F_k(r, i, \text{None}))$;
      r_score_len $\leftarrow$ r_score_len $+ 1$;
    **else**
      // Correct $r$ found
      score_len $\leftarrow$ score_len $+ 1$;
      **for** $\sigma \in \{0, 1, \leftarrow\}$ **do**
        score$_\sigma \leftarrow$ score$_\sigma + s(r, x_i, F_k(r, i, \sigma))$;
        **if** $(score_\sigma - score\_len)/\sqrt{score\_len} > t$ **then**
          code $\leftarrow$ code $|| (\sigma)$;
          score$_\sigma \leftarrow 0$ for $\sigma \in \{0, 1, \leftarrow\}$;
          score_len $\leftarrow 0$;
          **break**;
        **end**
      **end**
    **end**
  **end**
  **if** $code \neq ()$ **then**
    **return** decode(code);
  **end**
**end**
**return** False;

---

