# OpenReview forum: "Undetectable Steganography for Language Models"
_TMLR — Accepted by TMLR_

### Review · Reviewer_2zmr · 2024-07-17

**Summary Of Contributions:**

This manuscript proposes a cryptographic method that allows to encode a payload into outputs generated by popular large language models and decode it by means of a linked method that makes use of a secret key, also employed in the encoding part. This work builds on the previous work by Christ, Gunn and Zamir by extending watermarking from true/false to a string with length that appears to be linear in the lengths of the response. The key technical approach is a dynamic variant of error correcting codes with feedback. The manuscript also includes demonstration of encoding payload "JD" into 150+ word response generated by GPT-2 following 20+ word query.

**Audience:**

Yes

**Claims And Evidence:**

Yes

**Requested Changes:**

Most of changes I believe should take place are of major nature, which perhaps are not suitable for this kind of peer-review process. These are:

  A. Significant reduction of background material or complete withdrawal of it to appendices (sections 2 and 3)

  B. Adding line numbers to all algorithms and referencing them in writing

  C. significant revision of sections 4, 5 and 6 which includes

    i) replacing high-level description followed by low-level description as done in the manuscript with a single, self-contained, description of the proposed method

    ii) adding description of dynamic ECC with feedback into the main body as one of the key contributions

Some smaller changes are:

  D. Word "arbitrary" as used in this manuscript is not specific enough and contradicts what you say in the manuscript (the manuscript states that there is linear relationship between response and payload which contradicts the arbitrariness of the payload)

  E. Word "rigour" as used in this manuscript on multiple occasions suggests self-assessment. I would suggest you remove it and let reviewers and audience decide on the rigorousness.

  F. Numerous typos/grammar issues (can be extend, this is length)

  G. page 4 "takes as input" - what?

  H. page 5 talks about entropy you require but do not discuss and present evidence of typical entropies exhibited by LLMs

  I. page 6 mentions F_{k}(r,i) but I don't believe you ever defined it

  J. section 4 really needs a diagram to follow your train of thought

  K. page 11 refers to overhead as straightforward - what does it mean?

  L. section 8, would not a restart require lengthening of the response, can you discuss that?

**Strengths And Weaknesses:**

Strengths

  A. This is a practically relevant problem and the proposed solution is an important step in this direction.

  B. The dynamic ECC with feedback may be of interest in other areas

Weaknesses

  C. Manuscript

    i) A very substantial part of the manuscript largely repeats Christ, Gunn and Zamir. In a setting where you are limited by the number of pages but perhaps less so by the size of appendices this choice is peculiar.

    ii) None of the four algorithms included has line numbers which affects the quality of their integration with text.

    iii) Section 4 is ~2 pages of pure text that forward references "rigorous construction" in sections 5 and 6, and back references section 4. There is no attempt made here to refer to specific parts of some algorithms or present a diagram.

    iv) Section 5 which includes perhaps the key technical contribution is a sentence that refers to information in the appendix.

    v) Repeated use of high-level followed by somewhat low-level descriptions is distracting. Repeated forward references are distracting. A single self-contained piece of writing would be a better choice.

  D. Empirical investigation

    i) Very limited empirical investigation

    ii) It appears that although linear the proposed method enables encoding only a limited number of bits of information. There is not much discussion on that and implications given that the length of LLM responses is not a free variable.

---

> ### Author Response · Authors · 2024-07-23
> **Reply to Reviewer**
>
> Dear Reviewer,
>
> We thank you very much for the thorough review.
> We will respond next to the questions/weaknesses raised in the review:
>
> 1. Empirical Investigation:
> In the paper we prove that the amount of information we can hide in a response is linear in the entropy of the generation process (which is best possible, up to constants).
> We then point out (discussed in Section 2.3.1) that studies in Linguistics and previous research on LLMs imply that the entropy of natural text grows linearly with its length (which is the best possible scenario, up to constant factors).
> Thus, the empirical investigation's objective is only to demonstrate that the second part (with regards to the entropy in SOTA language models being linear) is indeed true, and that the constants 'suffered' by our construction aren't too outlandish: We show that we can indeed hide a number of bits linear in the response length.
> As this study is theoretical in nature, we did not optimize those constants further - we believe that significant optimization is possible (but this would not be appropriate in this work as it wouldn't improve the theoretical bounds).
>
> 2. Organization of the Writeup:
> We chose to use the (rather short) main body of the paper for giving the complete set of definitions and theorem statements, and only a high-level sketch of the construction itself - deferring the technical proofs to the appendix.
> We wanted this sketch to be self-contained and thus included within it also a sketch of the relevant parts of CGZ we built upon.
> We understand that while we view this format as more friendly to the reader, this view is subjective and other readers might prefer having no high-level sketch preceding the actual proofs.
> In a "camera-ready version", we would take into account the concrete comments and suggestions mentioned in the review that fit with this paper organization.
>
> Replies to Questions in the "Requested Changes" Part:
>
> H. The empirical entropy of natural languages is discussed in Section 2.3.1.
>
> K. This is a typo (a mix of previous versions saying that the run-time overhead is minimal and that the implementation of the change to the inference-algorithm is straightforward), will be fixed.
>
> L. Yes. For entropy reasons, you can't fit more than m bits of information in a process with entropy < m. On the other hand, if you are trying to convey a small number m of information bits in a much longer response (of length n>>m), then by "restarting" the algorithm after each time you manage to encode m bits, you'd end up with a response that encodes the m-bit payload n/m times, and each of them can be recovered independently - giving you some robustness to edits/cropping.
>
> All other typos/style issues mentioned will be fixed.
>
> We will be happy to answer further questions.
>
> Thanks.

---

### Review · Reviewer_DHD8 · 2024-08-16

**Summary Of Contributions:**

This is a nice theoretical paper, where the authors propose a method to hide an arbitrary message in the output of an LLM. The hidden message can be extracted from the output using a secret key. The key property of this method is that without this secret key, the outputs of the original LLM and the modified LLM hiding the message are provably indistinguishable, and therefore does not affect the quality of the generated outputs. The work extends an undetectable watermarking scheme for LLMs proposed by Christ et al (2023), such that arbitrary messages can be hidden in the output of an LLM. Overall, the paper makes good theoretical progress towards solving an important problem, and I believe should be accepted.

**Audience:**

Yes

**Broader Impact Concerns:**

No concerns.

**Claims And Evidence:**

Yes

**Requested Changes:**

It would be great if the authors could refer to the Weaknesses section above and resolve some of those issues, especially the comments on improving the writing and adding further discussion.

**Strengths And Weaknesses:**

Strengths

- One of the main strengths of this work is in extending the work of Christ et al (2023) to obtain a method that can hide an arbitrary secret message in the output of an LLM while keeping the output distribution of the LLM unchanged. The outputs of the modified LLM hiding the message is indistinguishable from the original LLM without access to a secret key.

- To apply the undetectable watermarking scheme of Christ et al (2023) for the purpose of hiding arbitrary messages in the LLM output, the authors take two important steps. 1) By breaking up the message into smaller parts and encoding each part separately, the authors are able to keep the running time of the retrieval algorithm of the method low. 2) By using Error Correcting Codes, the authors are able to extend the size of the hidden message to be optimal with respect to the size of the generated text.

- The method proposed has provable guarantees, both in terms of indistinguishability of the outputs between the modified and the original LLMs, as well as showing that the retrieval algorithm can retrieve the hidden message with reasonable assumptions on the entropy of the generated response.

- The authors are transparent about the limitation of the method, which is not robust to edits of the generated text.


Weaknesses

- Experiments are a bit lacking. The only experiment in the paper is on showing that the length of the hidden message grows linearly with the length of the LLM output when generating natural language. However, even this experiment is only shown for 5 data points and a maximum output length of only 100 tokens, which is very short compared to practical use cases of LLMs.

- There is no discussion on the practical implementation of the proposed method. While an algorithm is proposed and the authors analyze its theoretical complexity, there is no discussion on the practical runtimes of the algorithm. While Figure 3 seems to be going through the decoding process, more discussion on this would be great, including runtimes. There is also no discussion on ways to check the correctness of the implementation, which would likely be crucial for the practical security of the proposed method.

- Not much discussion of prior work on hiding arbitrary messages in the output of an LLM. While some of these works are cited, it would be great to have a more thorough comparison with these earlier works.

- The writing could be improved. In particular, the part around the dynamic Error Correcting Codes. While this seems like a key contribution of the paper, it is not properly explained in the main paper, and the short paragraph on it in Section 4 is a bit unclear. There are also a number of small typos in the paper that should be fixed.

---

> ### Author Response · Authors · 2024-08-20
>
> Thank you very much for your detailed review!
>
> We will reply to the questions asked:
> 1. Practical implementation: The full pseudo-code for the implementation appears in Algorithms 3-6 in the paper. Furthermore, the code implementing them (using any model from HuggingFace) which we used for the experiments appears publicly on GitHub; We did not include the link in the version for review due to its double-blind nature. It appears in the non-anonymous version.
> 2. We will add some more discussion in the main part of the paper about the parts that are currently in the introduction, including the Dynamic ECC.
> 3. Experiments: We included a public (GitHub) implementation and ran basic experiments for two reasons; The first, is point (1) above - showing that the practical implementation is flexible to work with any LLM and rather simple; The second, is showing that as implied by the studies in Linguistics we cite and discuss, it is indeed the case with modern LLMs that the entropy appears to grow linearly with the length of the response. Nonetheless, the paper is theoretical in nature and those experiments are only meant to complement the proofs.
>
> We will discuss those points in the final version of the paper, to clarify. We will be happy to answer further questions.

---

### Review · Reviewer_bPsq · 2024-09-05

**Summary Of Contributions:**

This paper demonstrates how a watermarking scheme for LLMs proposed by Christ, Gunn & Zamir can be extended in order to implement a stenographic scheme that encodes a hidden message in the output of the model which can be retrieved with knowledge of the secret key. Like the CGZ scheme, this is done without changing the distribution of model outputs, and so is theoretically undetectable.
The authors also demonstrate a practical implementation of their scheme on GPT-2 and Llama 2, demonstrating that it can encode non-trivial messages in reasonable lengths of outputs.

In many ways this paper is unsurprising, as (as the authors point out) any watermarking scheme can be converted into a stenographic one. However, the details of doing this in a practical way, particularly the use of error correcting codes to allow the encoder and reciever to co-ordinate on the locations of the ‘blocks’ of the message to be decoded without the ability to co-ordinate, together with the practical demonstration of the technique, are easily substantial enough, in my view, for the paper to be a valuable contribution to the literature in it’s own right.

The paper briefly describes the CGZ watermarking scheme, which essentially allows the reliable decoding of 1 bit of information (was the text generated by the watermarked model?) from the text generated by a model, given knowledge of the secret key. This is then extended to allow transmission of multiple bits of information, allowing the transmission of a multi-bit payload which can be decoded with knowledge of the secret key. The main technical issues to be solved are that a successful decoding requires sufficient empirical entropy in the generated message, which is not known in advance given the setting, and the requirement to ‘pad’ the start of the message with genuine random decoding in order to avoid deterministic generation which could reveal the existence of the decoding scheme to an adversary who allowed multiple queries. The authors describe how these can be resolved in practice.

**Audience:**

Yes

**Broader Impact Concerns:**

I don't think that a broader impact statement is required.

**Claims And Evidence:**

Yes

**Requested Changes:**

I think that the paper is probably mostly suitable for acceptance in it’s current form, but I would be interested to see discussion of the following points:

The paper conducts some experiments on Llama 2, the open-weights model released by Meta. This model is consistently referred to as ‘llamma’ instead: I think that this is a mistake and should be fixed?

I would be interested in a bit more discussion of possible applications and the requirements for this. For instance, the authors suggest in the introduction that it would be possible to encode ‘the users name, the time, or the prompt used’ into the output of the message; but in practice this would likely be difficult. For instance, if a company has 10 million users, then this requires around 24 bits of information in order to uniquely identify them all, but the authors empirical results seem to suggest that this requires a fairly long message (> 100 tokens) in order to encode reliably while remaining undetectable. Possibly the constants could be improved but I think a bit more discussion on this could be illuminating.

One potential factor which I thought could be an interesting extension of this paper is the assumption that the model is a deterministic mapping from inputs to outputs (definition 2.1). Many deployed models are not fully deterministic in practice, either due to hardware indeterminism or batch effects with mixture of expert models (or a combination of both). These deviations from full determinism could probably be modelled as small amounts of noise : would this affect the feasibility of the error correction scheme? Intuitively I feel like the method could be modified to cope with this (as it’s basically a very slightly noisy channel), which might be both technically interesting and increase the immediate applicability of the method. As mentioned above, though, I think that the paper is sufficiently interesting as a contribution in it’s current form.

**Strengths And Weaknesses:**

Strengths: the paper is well written and the exposition of the main technique is clear. The assumptions and difficulty of the method is explained well. The empirical evaluation is valuable, and gives some indications that the method could be useful in practice.

Weaknesses: The paper is mostly an extension of an existing technique, and makes frequent references to the paper on which it builds. The practical implications of the technique are not discussed in great detail. Some of the potential applications mentioned in the introduction are not necessarily well supported in practice; for example, the introduction mentions the possibility that, rather than simply being able to detect that a text was generated by a particular LLM, a stenographic scheme would allow the determination of the identity of the user and the time of generation. However, the empirical results (figure 2) seem to suggest that, while the length of the encodable payload grows linearly with the number of generated tokens as the author predict, in practice the encodable payloads are quite short unless the message is fairly long (it seems to be that for N tokens, the scheme is capable of encoding around N / 5  bits of information, which is not a lot given the average length of an LLM message.)

---

> ### Author Response · Authors · 2024-09-05
>
> Thank you very much for the thorough review!
>
> 1. Length of the payload:
> The paper is theoretical in nature, and the main theoretical result is that the amount of bits you can encode grows linearly with the empirical entropy (or equivalently in natural models, the length of the response). It could have been a-priori that the capacity has to depend on the security parameter or grows significantly slower when the response gets larger. This result is the theoretically best-possible (up to a constant factor) as you can never encode more information in a channel than its entropy.
> The point we attempt to get across with the experimental part, is that the theoretical bounds reasonably translate to practice when the same scheme is implemented - that is, we don't ``hide" very large constants, etc.; On the other hand, we do not attempt to optimize the constants further in the implementation and do not explore additional heuristics that would likely help in practice.
> While without further optimization we indeed don't have enough capacity to encode long metadata, we pose those scenarios as objectives (or concerns) that should be viewed as highly plausible given the current results - which we see as an academic "proof of concept" rather than an optimized "final product".
>
> 2. Non-determinism of the model:
> The only property we truly use (as did CGZ) is that when you generate a response you generate it token-by-token, and that for each token you generate you draw it out of an explicitly written distribution over all possible tokens. As long as you remain within this framework (and as far as we understand you do) then determinism is not necessary. Thus in fact you don't even need to change the proofs or add noise for it to work.
>
>
> Other comments: We will fix the references to be "Llama 2" instead of "Llama".
>
> We will be happy to answer further questions.
>
> Thanks.

---

> > ### Comment · Reviewer_bPsq · 2024-09-23
> > **further response**
> >
> > So on point 2. I'm not quite sure that this is right; as I understand the model as a mapping from prefix to distribution over the next token (i.e p(y) = model(X) where X is a prefix and y is the next token, and assume that this is deterministic. In the cases I mentioned, model(X) is not a deterministic mapping (i.e the p you get isn't purely a function of X, but can also depend on both noise and other batch elements etc.). From your reply, maybe you don't actually need model to be deterministic provided you have explicit access to p(y)? (I don't really have time to read the paper again to improve my understanding).
> >
> > In that case adding a brief sentence/footnote to the paper explaining this point might be of interest to those thinking of applying it in industry, since as I say determinism of the forward pass is frequently violated in practice.

---

> > > ### Author Response · Authors · 2024-09-23
> > >
> > > We will add a sentence to make it clear. Essentially, the point here is that on detection time there is no access to the prompt (you only see the response when you detect the watermark/steganography - not what the prompt was) and in particular there is no access to the model at all (as it could be that the prompt describes which model to use, and when you don't have the prompt you also don't have the model's distributions); Hence, as long as there is some explicit token distribution used at each step it does not matter if it was fully determined by the (unknown) prompt and prefix, as any additional 'decisions' can be viewed as part of the unknown prompt.

---

### Decision · Action_Editor_oNZE · 2024-10-12

**Recommendation:** Accept as is

**Comment:**

As stated above, this is a simple ideal which is well addressed experimentally and well explained. The reviewers concurred that the paper was publishable, but felt it did not have enough unique content to automatically progress to ICLR. It was suggested that a joint submission with the original watermarking work, if that work had author overlap with this one, would make for a strong submission.

**Audience:**

This should be of pretty general interest to both people interested in theoretical properties of LLMs, and applications of LLMs.

**Claims And Evidence:**

This paper explains how watermarking methods, used to identify a particular model's outputs as being from that model, can be extended to the encoding of arbitrary information in an theoretically undetectable manner (i.e. steganography). The idea may sound simple but the practical implementation and application is described and evaluated in this paper, which is where the value lies.